# High-fidelity DNA ligation enforces accurate Okazaki fragment maturation during DNA replication

Jessica S. Williams[1,2], Percy P. Tumbale[1,2], Mercedes E. Arana[1,2], Julian A. Rana[1], R. Scott Williams ● [1✉] & Thomas A. Kunkel ● [1✉]

DNA ligase 1 (LIG1, Cdc9 in yeast) finalizes eukaryotic nuclear DNA replication by sealing Okazaki fragments using DNA end-joining reactions that strongly discriminate against incorrectly paired DNA substrates. Whether intrinsic ligation fidelity contributes to the accuracy of replication of the nuclear genome is unknown. Here, we show that an engineered low-fidelity LIG1[Cdc9] variant confers a novel mutator phenotype in yeast typified by the accumulation of single base insertion mutations in homonucleotide runs. The rate at which these additions are generated increases upon concomitant inactivation of DNA mismatch repair, or by inactivation of the Fen1[Rad27] Okazaki fragment maturation (OFM) nuclease. Biochemical and structural data establish that LIG1[Cdc9] normally avoids erroneous ligation of DNA polymerase slippage products, and this protection is compromised by mutation of a LIG1[Cdc9] high-fidelity metal binding site. Collectively, our data indicate that high-fidelity DNA ligation is required to prevent insertion mutations, and that this may be particularly critical following strand displacement synthesis during the completion of OFM.

[1] Genome Integrity and Structural Biology Laboratory, National Institute of Environmental Health Sciences, US National Institutes of Health, Department of Health and Human Services, 111 TW Alexander Drive, Research Triangle Park, NC 27709, USA. [2] These authors contributed equally: Jessica S. Williams, Percy P. Tumbale, Mercedes E. Arana. ✉email: williamsrs@niehs.nih.gov; kunkel@niehs.nih.gov

The most abundant DNA transaction in eukaryotic cells that affects the fidelity of nuclear DNA replication is the nucleotide selectivity of DNA polymerases (Pols) α, δ, and ε during incorporation into a growing DNA chain during leading and lagging strand synthesis[1]. Second only to this selectivity is proper Okazaki fragment maturation (OFM) of lagging strand DNA fragments, an event that occurs thousands to millions of times during each DNA replication cycle in eukaryotic cells[2–4]. Okazaki fragment processing involves replacement of RNA–DNA primers made by Pol α-primase. This occurs by nick-translation/ strand displacement DNA synthesis catalyzed by Pol δ to create DNA substrates that are cleaved by Fen1 and/or by DNA2 nucleases[5–10]. The resulting nick is then sealed by DNA ligase 1 (LIG1) to finalize synthesis of the DNA strand. Successful OFM produces undamaged DNA ends that are sealed by LIG1 to ensure faithful DNA replication and avoid genome instability in the form of DNA breaks, deletions, and duplications.

Among four known eukaryotic DNA ligases[11–13], LIG1 is the enzyme involved in OFM. A recently published high-resolution structure of LIG1 bound to a nicked DNA duplex reveals that it employs a $Mg^{2+}$-reinforced DNA-binding mode to promote high-fidelity DNA ligation in vitro[14]. The interface formed between LIG1, $Mg^{2+}$, and the DNA substrate is critical for ensuring that DNA nicks containing damaged or mutagenic ends are not sealed, thereby promoting high-fidelity DNA ligation. Mutation of two highly conserved $Mg^{2+}$-binding glutamic acid residues in LIG1 that form this interface to alanines reduces enzyme accuracy but does not impact enzymatic turnover, thereby facilitating mutagenic ligation[14]. However, despite the importance of DNA ligation reactions for life, whether high-fidelity DNA ligation is required for faithful DNA replication in vivo remains unknown.

Here we demonstrate that LIG1 is a highly accurate DNA ligase in vivo. Expression of a Cdc9$^{EE/AA}$ variant in yeast that compromises ligation fidelity results in a high rate of +1 base insertion events in short homonucleotide runs. These mutagenic events are exacerbated upon loss of DNA mismatch repair (MMR) or loss of Fen1-dependent nuclease activity, demonstrating that highly accurate DNA ligation is a previously underappreciated critical determinant of faithful replication of the nuclear genome. Biochemical and structural dissection of the mutagenic ligation reaction performed by the low-fidelity mutant DNA ligase provides a molecular framework for this insertion mutagenesis and solidifies the importance of accurate DNA ligation during DNA synthesis.

## Results

**The in vivo mutagenic consequences of low-fidelity DNA ligation.** To test whether high-fidelity ligation is critical for genome maintenance in vivo, we constructed a low-fidelity ligation yeast strain containing alanine substitutions of the conserved Glu206 and Glu443 residues in the *CDC9* gene that encodes *Saccharomyces cerevisiae* DNA ligase 1 (*cdc9-EE/AA*) (Fig. 1a). Tetrad dissection reveals that spore colonies harboring the mutant *cdc9*-EE/AA ligase germinated and grew to a colony size similar to a wild-type control (Supplementary Fig. 1). Using the *URA3* forward mutation reporter assay, we measured spontaneous mutation rates and specificity in the *cdc9-EE/AA* mutant. An overall increase (2.8-fold) in mutation rate compared to a wild-type (*wt*) yeast strain (Fig. 1b) was driven almost entirely by a significant number of single base addition mutations that were observed (Fig. 1c). These single base insertions are within short homonucleotide runs, predominantly in runs of G–C base pairs but occasionally in runs of A–T base pairs (Fig. 1c), suggesting that formation of the +1 additions likely involves DNA strand

slippage. The specificity is also intriguing in that the single base additions observed here, whose production requires an extra base in the newly synthesized strand, are much more prevalent than are the single base deletions that predominate in many other studies[15–21], and require an extra base in the template strand to generate a misaligned intermediate with an unpaired nucleotide[22,23]. Moreover, the observed insertions are spaced intermittently, but non-randomly, throughout the *URA3* reporter gene. Thus, despite the presence of homonucleotide runs at many locations in *URA3*, the addition mutations are primarily observed at locations separated by approximately 200 base pairs (Fig. 1c, positions 157, 344, 564, for example). Interestingly, high-resolution mapping of Okazaki fragments from *S. cerevisiae* showed that their size is roughly consistent with the nucleosome repeat, which averages 165 bp in size[24,25], and DNA ligase 1 is able to efficiently seal a nick on a nucleosomal substrate[26]. The mutation rates for +1 insertions are >300-fold increased relative to wt for *cdc9-EE/AA* (Fig. 1d). In marked contrast, single base substitutions and deletion mutations are infrequent in *cdc9-EE/ AA* (Fig. 1c, f), as are mutation events involving multiple bases (Supplementary Table 2).

We next examined the role of DNA MMR in correcting these +1 insertion mutations observed in the *cdc9-EE/AA* mutant by deleting each of the three *MSH* genes (*MSH2*, *MSH3*, or *MSH6*). Loss of these genes in the low-fidelity ligase mutant had a modest effect on overall mutation rate, with the largest effect observed in the *cdc9-EE/AA msh2Δ* strain (Fig. 1e). Msh2 is the common component of the MutSα (Msh2-Msh6) and the MutSβ (Msh2-Msh3) heterodimers that initiate MMR, and therefore loss of *MSH2* eliminates MMR-dependent repair of single base substitutions, as well as small and large deletion and insertion mutations[20,27]. Here we observe a fourfold increase in overall mutation rate when comparing the *cdc9-EE/AA msh2Δ* double mutant to the *msh2Δ* strain (Fig. 1e). These mutator effects encouraged us to sequence *ura3* mutants to determine whether specific classes of mutations are affected in the absence of MMR.

The +1 insertion rates are further elevated in *cdc9-EE/AA* MMR-deficient strains, despite *cdc9-EE/AA* having very little effect on the rate of single base substitutions or deletions when compared to the *msh6Δ*, *msh3Δ*, or *msh2Δ* single mutant strains expressing wild-type *CDC9* (Fig. 1f and Supplementary Tables 3 and 4). The relative increase in the +1 insertion rate is greatest in the *cdc9-EE/AA msh2Δ* mutant strain (Fig. 1d and Supplementary Fig. 2, ~80-fold higher compared to *cdc9-EE/AA*). The slightly smaller sizes of the *cdc9-EE/AA msh2Δ* haploid strain colonies, as well as the decreased growth rate and enlarged cells (Supplementary Fig. 1), may be related to the observed high rate of the addition mutations, suggesting that these *cdc9-EE/AA msh2Δ* mutant cells may have accumulated +1 mutations in genes that are important for promoting normal cell growth and morphology. The +1 mutation rate is also strongly elevated (by 20-fold) in the *cdc9-EE/AA msh6Δ* mutant and is modestly elevated (by 1.5-fold) in the *cdc9-EE/AA msh3Δ* mutant when compared to the MMR-proficient *cdc9-EE/AA* strain (Supplementary Figs. 3 and 4 and Supplementary Tables 3 and 4). Quantitatively, these results are consistent with the interpretation that the Cdc9-dependent +1 mutations result from DNA replication errors, where loss of *MSH2* and *MSH6* have a much greater effect on formation of single base insertion/deletion (indel) errors than does a mutation in *MSH3*[20,28]. These facts, and the observation that the insertions occur at widely spaced locations in *URA3*, despite the presence of homonucleotide runs throughout the gene, are consistent with the role of MMR in reducing the rate of +1 insertions that are ligated during OFM in the low-fidelity *cdc9-EE/AA* mutant strain.

To further test the hypothesis that DNA ligase 1 performs high-fidelity ligation during replication, we deleted *RAD27* in the

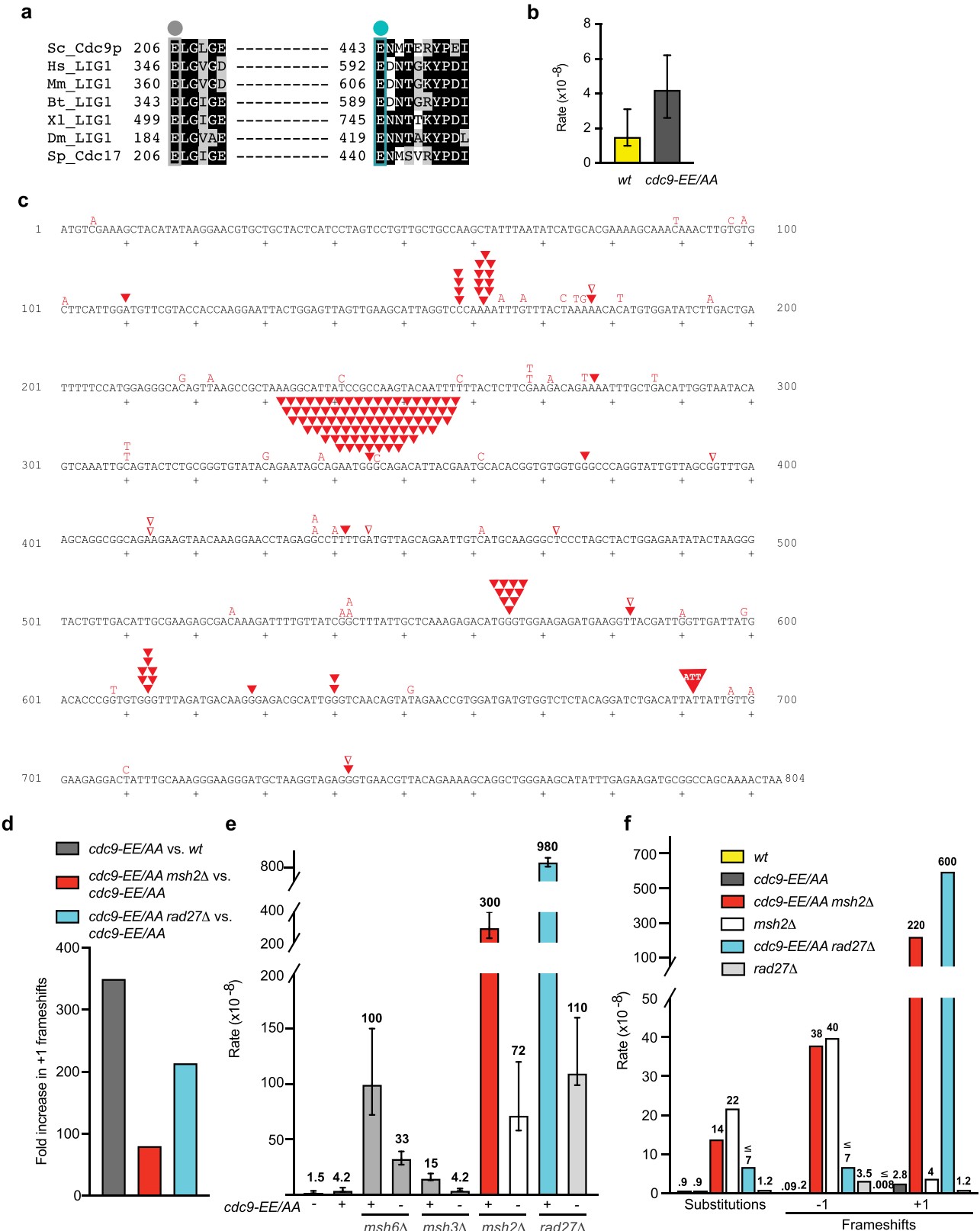

*cdc9-EE/AA* strain. *RAD27* is the gene encoding the Fen1 endonuclease involved in cleaving the displaced DNA flap generated by Pol δ displacement synthesis during OFM. Like the *cdc9-EE/AA msh2Δ* double mutant, the *cdc9-EE/AA rad27Δ*

haploid has a reduced spore colony size, a defect in cell growth, and enlarged cell morphology (Supplementary Fig. 1), again suggesting a modest growth disadvantage. The *cdc9-EE/AA rad27Δ* mutant also has a stronger overall mutator effect than

**Fig. 1 Mutational analysis of the low-fidelity *Cdc9-EE/AA* variant demonstrates the importance of Ligase 1 in preventing +1 insertion mutations during replication. a** Sequence alignment of the strictly conserved LIG1 HiFi glutamate residues in eukaryotic homologs. Sc *Saccharomyces cerevisiae*, Hs *Homo sapiens*, Mm *Mus musculus*, Bt *Bacillus thuringiensis*, Xl *Xenopus laevis*, Dm *Drosophila melanogaster*, Sp *Schizosaccharomyces pombe*. **b** Overall spontaneous mutation rates for yeast strains expressing either wild-type Cdc9 or the Cdc9-EE/AA variant were determined by fluctuation analysis using a *URA3* reporter assay (biological replicates: n = 34 for wt; n = 52 for *cdc9-EE/AA*). The rate of mutation conferring resistance to 5-FOA was determined as *ura3* mutants are resistant to 5-FOA. The median rate ± the 95% confidence interval is displayed. **c** The coding strand of the 804-bp *URA3* gene is shown. Small sequence changes (≤3 bp) observed in independent *ura3* mutants are depicted in red for the *cdc9-EE/AA* strain (n = 189). Letters indicate single base substitutions, inverted closed triangles indicate +1 frameshifts, and inverted open triangles indicate −1 frameshifts. Larger sequence changes observed are listed in Supplementary Table 2. **d** The fold increases in the +1 frameshift mutation rate for the indicated comparisons were calculated using the specific mutation rates displayed in **f**. **e** Overall spontaneous mutation rates for the *wt* or *cdc9-EE/AA* strains +/− MMR or RAD27 are displayed as in **b** (biological replicates: n = 39 for msh6Δ; n = 33 for *cdc9-EE/AA* msh6Δ; n = 22 for msh3Δ; n = 48 for *cdc9-EE/AA* msh3Δ; n = 47 for msh2Δ; n = 81 for *cdc9-EE/AA* msh2Δ; n = 97 for rad27Δ; n = 261 for *cdc9-EE/AAA* rad27Δ). **f** The specific mutation rates of single base substitutions, −1 frameshifts, and +1 frameshifts were calculated as the proportion of each type of event among the total mutants sequenced (Supplementary Table 3), multiplied by the overall mutation rate (**e**). Specific mutation rate values are indicated above each bar, and a ≤ symbol indicates that there were 0 events observed. The sequencing data for wt and msh2Δ is from ref. [52]. Source data for panels are provided as a Source data file.

either the *cdc9-EE/AA* or the *rad27Δ* single mutants (Fig. 1e). More importantly, sequence analysis of independent *ura3* mutants demonstrates that, in addition to the well-known duplications of multiple base pairs seen here (Supplementary Tables 5 and 6) and previously observed in *RAD27*-deficient strains (Supplementary Table 7)[29,30], the *cdc9-EE/AA rad27Δ* double mutant generates +1 insertions (Supplementary Fig. 5) at a very high rate (Fig. 1f). The rate is elevated 200-fold when compared to the *cdc9-EE/AA* strain (Fig. 1d) and is synergistic with loss of Fen1-dependent OFM. These +1 mutations are again observed in homonucleotide runs that are non-uniformly distributed throughout the *URA3* reporter gene (Supplementary Fig. 5), with some of the +1 insertions occurring in the same homonucleotide runs observed in the MMR-defective strains (Supplementary Figs. 2–4). Others are in different locations, e.g., in runs of T–A base pairs at positions 90–93 and 440–443 (Supplementary Fig. 5). The differences in the specificity of +1 addition mutagenesis in the *cdc9-EE/AA* mutant lacking either *MSH2* or *RAD27* may be anticipated by the fact that +1 additions can result from defects in MMR or during OFM when DNA flaps are removed by Rad27, the Dna2 helicase-nuclease, or by both nucleases[8,10,31,32].

Based on error signatures[33,34] and ribonucleotide incorporation[35–38], it can be inferred which DNA strand is being synthesized as the lagging strand (in green in Fig. 2a) by Pol δ or the leading strand (in blue) by Pol ε during synthesis of the *URA3* reporter in its genomic location adjacent to the closest replication origin, *ARS306*. These studies were performed using mutator variants of DNA Pols δ and ε that had been biochemically characterized as having specific mutation signatures and ribonucleotide incorporation propensities. The contributions and the sequence contexts for four of the +1 C/G insertion hotspots in *URA3* observed in the *cdc9-EE/AA* mutant are displayed in Fig. 2a. Because of the discontinuous nature of lagging strand synthesis and the fact that DNA ligase 1 completes OFM by joining the DNA fragments, we can infer that it is an extra base incorporated by Pol δ during synthesis of the lagging strand that leads to error-prone ligation and a +1 insertion mutation signature in the *cdc9-EE/AA* strain. As a result, this would suggest that it is incorporation of an extra G at hotspot 157 or incorporation of an extra C at hotspots 344, 564, and 612 by Pol δ that generates the DNA substrate ligated by the low-fidelity mutant Cdc9 ligase (Fig. 2a). For two of the hotspots, the +1 insertion mutation rates are elevated by 240-fold (position 344) and 25-fold (position 564) when comparing the *cdc9-EE/AA* mutant to *wt* (Fig. 2b and Supplementary Table 8). These rates are further elevated in the absence of either *MSH2* (45-fold and 100-fold) or *RAD27* (79-fold and 750-fold), suggesting that MMR, Fen1, and high-fidelity DNA ligation are critical for

**a**

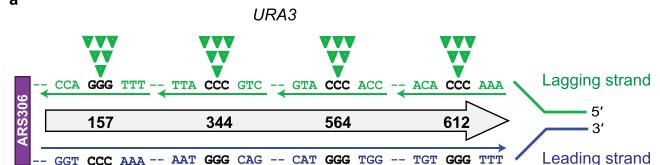

**b**

| Genotype | Mutation rate (x10⁻⁸) of C/G insertions at *URA3* position | | | |
|---|---|---|---|---|
| | 157 | 344 | 564 | 612 |
| wt | ≤0.0079 | ≤0.0079 | ≤0.0079 | ≤0.0079 |
| *cdc9-EE/AA* | 0.088 | 1.9 | 0.2 | 0.15 |
| *cdc9-EE/AA msh2Δ* | 37 | 86 | 20 | 33 |
| *msh2Δ* | ≤0.4 | ≤0.4 | ≤0.4 | ≤0.4 |
| *cdc9-EE/AA rad27Δ* | 7.0 | 150 | 150 | 14 |
| *rad27Δ* | ≤1.2 | ≤1.2 | ≤1.2 | ≤1.2 |

**c**

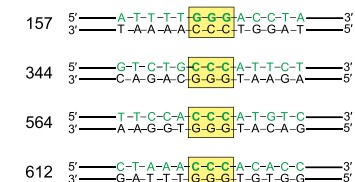

**Fig. 2 Analysis of mutational hotspots for +1 insertions in the low-fidelity *cdc9-EE/AA* strain +/− *MSH2* or *RAD27*. a** A schematic of the *URA3* reporter gene located adjacent to the *ARS306* replication origin. Displayed is the leading strand DNA sequence synthesized by Pol ε in blue and the lagging strand synthesized by Pol δ in green. The additional C that would be incorporated by Pol δ at three of the mutation hotspots to generate a +1 insertion is indicated by the closed green inverted triangles. **b** Site-specific rates for +1 G/C insertions at 4 hotspots in the *URA3* reporter gene for the *wt* and *cdc9-EE/AA* strains +/− *MSH2* or *RAD27*. Rates were calculated as the proportion of each site-specific event among the total mutants sequenced (Supplementary Table 3) multiplied by the overall mutation rate (Fig. 1c). **c** A depiction of the DNA sequences surrounding four of the sites in *URA3* at which the mutation rate of +1 insertions of G/C is the highest. The Pol δ-synthesized lagging strand is shown in green and the location of the homopolymer run in which an extra base would be inserted is highlighted by the yellow box.

preventing +1 insertions at defined genomic positions to maintain genome stability. The DNA sequence surrounding these *URA3* hotspots varies (Fig. 2c).

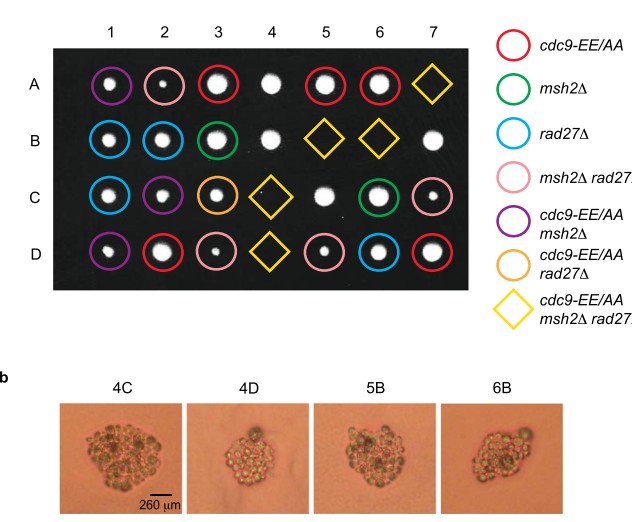

**Fig. 3 High-fidelity DNA ligation is required in the absence of both *MSH2* and *RAD27*. a** A diploid yeast strain heterozygous for *cdc9-EE/AA*, *msh2Δ*, and *rad27Δ* was sporulated, dissected, and the haploid spore colonies were genotyped by appropriate marker selection. 1–7 are tetrad dissections and A–D are haploid spore colonies. Plates were imaged after 3 days growth on YPDA at 30 °C. **b** Microscopic images of the indicated microcolonies taken after 5 days of growth reveals that the *cdc9-EE/AA msh2Δ rad27Δ* haploid derivatives sporulated but only divided a finite number of times. Thirteen dissections were performed using three independently derived diploid strains and similar results were obtained each time.

MMR was recently shown to act in an overlapping pathway with Rad27 to correct mutations that arise during OFM[39]. Therefore, we next tested the importance of high-fidelity ligation in the absence of both *RAD27* and *MSH2*. Tetrad dissection of a heterozygous *CDC9/cdc9-EE/AA*, *MSH2/msh2Δ*, and *RAD27/rad27Δ* diploid strain shows that the *cdc9-EE/AA msh2Δ rad27Δ* triple mutant haploid strain is inviable (Fig. 3a, yellow diamonds). Microscopic images of the spore colonies reveal that, following germination, the cells go through only a finite number of divisions before arrest (Fig. 3b) and never form colonies large enough to be macroscopically visualized (Fig. 3a). This demonstrates that high-fidelity DNA ligation by Cdc9 is critical for cell viability in the absence of both MMR and Fen1-dependent OFM.

**Mutagenic ligation of bulged DNA substrates**. We next probed the biochemical context under which Cdc9 can perform mutagenic ligation on substrates with single-nucleotide insertions. The lagging strand context of the *cdc9-EE/AA* mutator phenotype suggests that, following strand displacement DNA synthesis templated by triplet homonucleotide repeats (Fig. 4, step i), DNA strand slippage stabilizes a single bulged base (Fig. 4, step ii). DNA Pol extension of a bulged substrate (Fig. 4, step iii) precedes DNA flap maturation (Fig. 4, step iv). Cdc9 engagement of bulged substrates could occur in a variety of registers relative to the Pol misinsertion event, and therefore Cdc9 ligation activity was tested on an array of substrates where the DNA nick was placed at various positions relative to a bulged cytosine base (insC) within the *URA3* mutation hotspot sequence context observed in the *cdc9-EE/AA* strain (Fig. 5a).

Cdc9[WT]-catalyzed DNA ligation is markedly influenced by the positioning of a 1 nucleotide insC bulge (Fig. 5a, b, substrates insC1–insC8). Whereas Cdc9[WT] was active on all non-bulged substrate controls (Supplementary Fig. 6, substrates 1c–8c), it only ligated insC substrates with a bulge positioned 6–7

nucleotides 5′-terminal to a nick containing the ligatable 3′-hydroxyl (OH) and 5′-phosphate (P) ends (Fig. 5a, b, substrates insC1 and insC2). This result was expected, as the bulged cytosine nucleotide of insC1 and insC2 substrates falls outside the enzymatic footprint of eukaryotic DNA ligases on the upstream strand of a nick[14] and should therefore not impede catalysis (Supplementary Fig. 6). Strikingly, the low-fidelity Cdc9[EE/AA] enzyme could also ligate the insC4 substrate with a bulged nucleotide proximal to the nick (Fig. 5a, c, substrate insC4). Cdc9[WT] enzyme titrations (Fig. 5e, f and Supplementary Fig. 6d) show that Cdc9[WT] ligation activity is substantially impaired on insC4 compared to a non-bulged control, with >50% of catalytic events yielding abortive DNA ligation products (5′-AMP-modified DNA ends). In contrast, Cdc9[EE/AA] displays robust activity on insC4, with >70% catalytic events yielding mutagenic insertion ligation products (Fig. 5e, f) and modest accumulation (<5%) of abortive products.

Like for Cdc9[WT], under low-salt conditions (15 mM NaCl) human LIG1[WT] activity on bulged DNA is decreased overall and prone to production of abortive 5′-AMP intermediates (Supplementary Fig. 7). In higher salt (115 mM NaCl), LIG1[WT] displays no detectable catalytic activity on insC4, (Supplementary Fig. 7). Similar to the yeast enzyme, the marked discrimination of LIG1 against catalysis on insC4 is significantly suppressed for the low-fidelity LIG1[EE/AA] mutant, even in the higher stringency reaction conditions (Supplementary Fig. 7). We note that the bulged nucleotide could be accommodated in any of the 4 registers, including a 3′-flap conformation that is presumably non-ligatable. As a significant (~20%) fraction of the insC4 substrate remains unligated by the Cdc9[EE/AA] protein, even at our highest enzyme concentrations, variable conformations in annealing of the 3′ bulge may also impact ligation efficiency. Together, these results reveal that single-nucleotide bulges positioned 4 nt upstream of a DNA nick normally confound ligation by yeast and human LIG1 homologs, but low-fidelity Cdc9 and LIG1 variants efficiently circumvent this guard against abortive and mutagenic ligation.

**Molecular basis for mutagenic ligation by low-fidelity LIG1**. To define the basis for mutagenic ligation, we crystallized and determined X-ray crystal structures (Supplementary Table 9) of low-fidelity LIG1[EE/AA] bound to insC4 (2.8 Å resolution), a non-bulged 4c control (2.2 Å resolution), and compared them to the previously determined HiFi-Mg[2+]-bound LIG1[WT] DNA complex[14]. Overall, the DNA ligase structure adopts a closed, catalytically competent conformation poised for DNA nick sealing (step 3) and is characterized by DNA ligase DBD–AdD–OBD domain encirclement of the DNA substrate (Fig. 6a). The inserted bulged nucleotide is accommodated at the interface between the LIG1 DBD and AdD domains (Fig. 6b).

In principle, as insC4 has 4 C nucleotides that could be paired in variable registers with the 3 G nucleotides of the opposite strand, the bulge could be accommodated in any of three registers within the ligase active site. However, comparison of difference omit $F_o$–$F_c$ electron density for the upstream 3′-OH DNA strand reveals that a significant distortion in the DNA backbone path is observed for the insC4 sequence between the −3 and −4 positions relative to the DNA nick, indicating that the bulge is accommodated in a specific register (Fig. 6c). By comparison, fully paired duplex substrate is not bulged, as anticipated (Fig. 6d, "Methods", and Supplementary Movie 1).

Binding of the upstream 3′-OH strand by LIG1[WT] is normally reinforced by a protein–Mg[2+]–DNA interface, with Mg[2+]–HiFi metal coordination mediated by the stringently conserved Glu346 and Glu592 ligands (Fig. 6e). In a control mutant LIG1[EE/AA] unbulged DNA complex structure (Fig. 6f), a prominent pocket

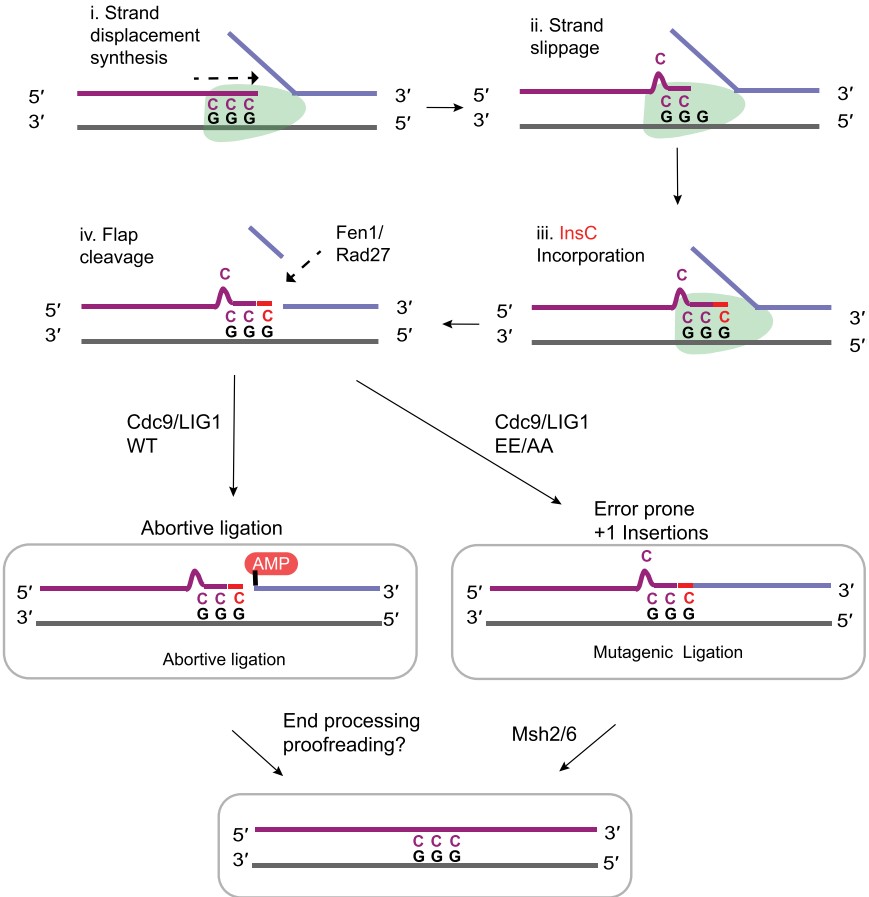

**Fig. 4 Cdc9 regulates the fidelity of Okazaki fragment maturation.** A model depicts a +1 base insertion mutation during OFM: (i) A DNA polymerase (green) performs strand displacement DNA synthesis templated by a triplet mononucleotide repeat. (ii) DNA strand slippage stabilizes a single bulged base. (iii) Polymerase extension of the bulged base results in a +1 base insertion in the new strand, (iv) Rad27/Fen1 cleaves the DNA flap. High-fidelity Cdc9$^{WT}$/LIG1$^{WT}$ aborts ligation on the bulged DNA to circumvent mutagenic ligation. Low-fidelity Cdc9$^{EE/AA}$/LIG$^{EE/AA}$ proceeds to mutagenic ligation.

created by the engineered EE/AA mutant is found at the DBD–AdD interface. It is this cavity which houses the bulged nucleotide in the insC4 complex (Fig. 6g). Notably, electron density for the budged nucleobase is less well defined than the backbone phosphates (Fig. 6c), indicating flexibility of the flipped nucleobase which is positioned near Pro341 and His337 in the EE/AA pocket. We conclude that structural plasticity imparted by the cavity created by deleting the HiFi metal and its ligands likely facilitates dynamic binding and flipping of the first C of the four cytosines (Fig. 6g).

## Discussion

Overall, our genetic and biochemical data demonstrate that high-fidelity in vivo ligation by the yeast replicative Cdc9 requires an intact high-fidelity protein–metal–DNA interface. High-fidelity ligation prevents sealing of DNA Pol-incorporated slippage mutations occurring at homopolymeric repeats, and expression of an engineered low-fidelity *cdc9-EE/AA* allele is a potent mutator in yeast. Biochemical studies of Cdc9 and a LIG1 X-ray structure with insC4 show how the mutant enzyme accommodates a bulged insC substrate by exploiting the engineered cavity created by deleting the high-fidelity metal-binding site in the enzyme.

The results presented here have several implications regarding OFM, the second most abundant DNA transaction occurring in eukaryotic cells. Overall, the data strongly imply that, under normal circumstances, wild-type LIG1 accurately seals nicks in newly replicated lagging strand DNA and that the fidelity of this

ligation reaction is strongly reduced by the Cdc9$^{EE/AA}$ mutation. In this study, the mutagenic effects of expressing the low-fidelity Cdc9$^{EE/AA}$ are largely devoted to +1 additions, but we cannot yet exclude that other mutations such as large multibase additions, duplications, or genomic rearrangements could result from suppressed ligation fidelity. A whole-genome approach using the low-fidelity *cdc9-EE/AA* mutant may also be useful for studying the role of DNA Pol α in replicating telomeric DNA[40] and/or the potential importance of high-fidelity ligation during OFM in preventing tumorigenesis. Strikingly, one of the mutational signatures identified in certain tumors is the addition of single base pairs in homonucleotide runs[41,42] that arose due to DNA Pol slippage during replication[23]. Could the +1 mutations in these tumors be caused by a defect in DNA ligation or other OFM-associated proteins?

The selective +1 error signature seen here has not been observed in studies of the fidelity of normal chain elongation by Family B DNA Pols α, δ, and ε[43]. Those DNA synthesis reactions by these Pols involve bulk chain elongation of open templates that lack an upstream duplex and, as such, do not require nick translation or strand displacement DNA synthesis[9,44]. The absence of a +1 error signature suggests that there may be fidelity protection mechanisms acting during nick translation/strand displacement synthesis characteristic of OFM. For example, is strand displacement synthesis energetically more costly than simple chain elongation, thereby resulting in DNA strand slippage that leads to single base additions when ligation is defective?

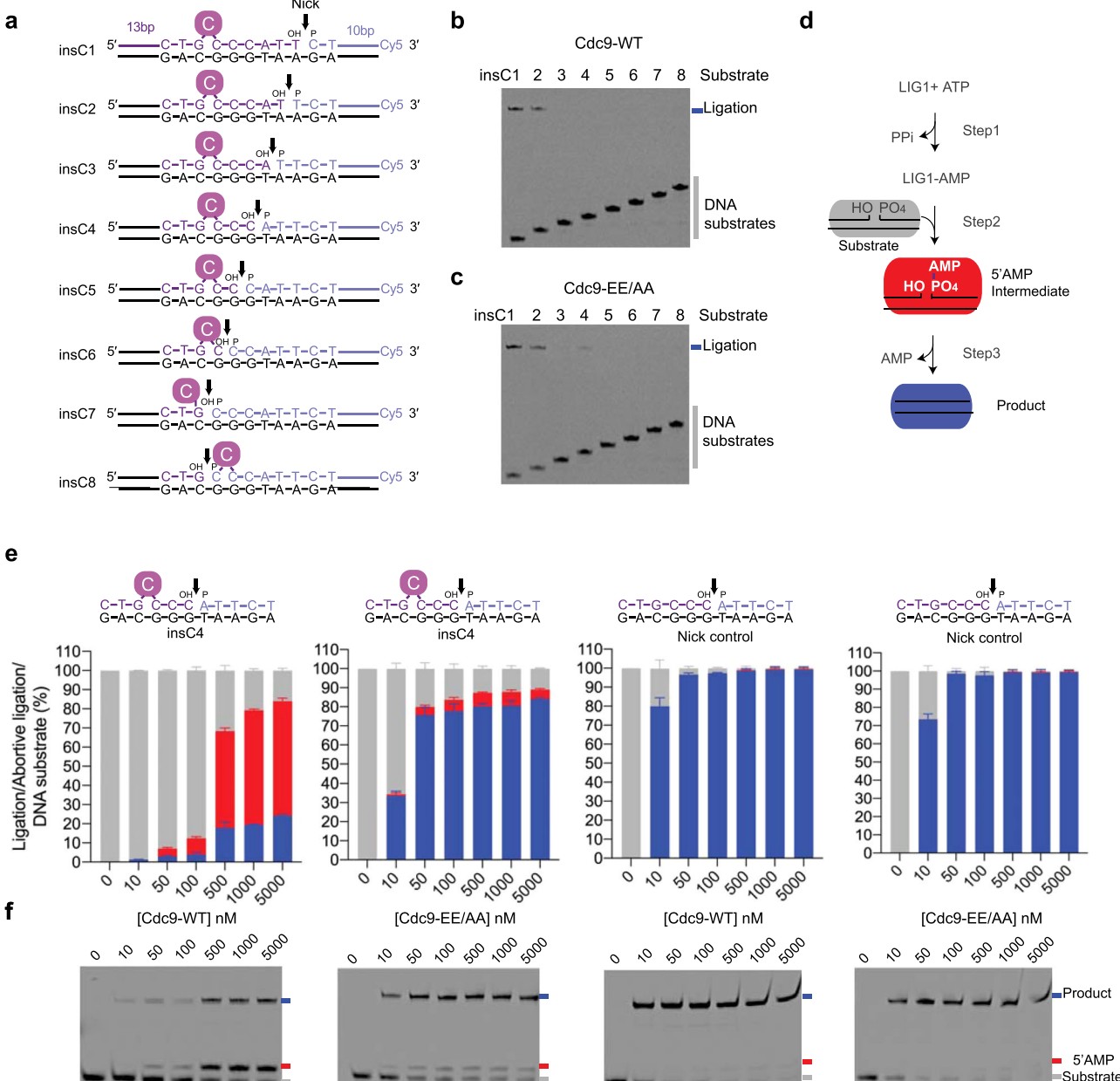

**Fig. 5 Cdc9 DNA ligation activity on bulged single-nucleotide insertion substrates. a** Schematic of the 3′-Cy5-labeled bulged insertion substrates (insC1–insC8). The DNA nicks (black arrows) of the bulged insertion substrates are placed at various positions relative to the bulged cytosine base (solid purple circle C) within the $cdc9^{EE/AA}$ URA3 reporter gene mutation hotspot sequence context. **b, c** Cdc9 ligation activity on insC substrates. A denaturing gel image of ligation reactions in the presence of 10 mM MgCl₂, 1 mM ATP, 15 mM NaCl, and 50 nM DNA substrate (insC1–insC8). Reactions contained 2 nM of Cdc9$^{WT}$ (**b**) or Cdc9$^{EE/AA}$ (**c**). Each of these experiments was repeated independently three times with similar results. **d** The three step ATP-dependent DNA ligation reaction. In Step 1, DNA ligase adenylates its active site lysine. The enzyme-AMP intermediate engages the nicked DNA substrate (gray) and the AMP group is then transferred to the 5′-phosphate of the nicked DNA to form the 5′AMP-DNA intermediate (red) in Step 2. A phosphodiester bond is formed in Step 3 to yield the ligated product (blue) and AMP is released. **e** Quantification of total catalytic events on insC4 (left 2 graphs) and control nicked DNA (right 2 graphs) producing ligated product (blue bars) and abortive DNA adenylation (red bars) by Cdc9$^{WT}$ or Cdc9$^{EE/AA}$ analyzed with the ImageQuant TL software (GE). Mean ± SD ($n = 3$ replicates) is displayed for 20 min ligation reactions. **f** Representative denaturing gel image of Cdc9 ligation reactions on insC4 (left 2 gels) or control nicked DNA (right 2 gels) in the presence of 10 mM MgCl₂, 1 mM ATP, 115 mM NaCl, and 50 nM DNA substrate. Reactions contained 0–5000 nM of Cdc9$^{WT}$ or Cdc9$^{EE/AA}$ at 25 °C for 20 min. Each of these experiments was repeated independently three times with similar results. Source data for panels are provided as a Source data file.

These results suggest that Pol δ makes +1 slippage mutations frequently during strand displacement synthesis, and we are currently testing whether the 3′-exonuclease activity of Pol δ is capable of proofreading such replication errors. DNA ligase prevents these mutations by acting as a molecular iron, suggesting that ligase fidelity is critical for preventing errors generated during strand displacement synthesis by Pol δ. While the current data suggest that the mutagenesis observed may be due to perturbation of OFM, they do not exclude other possibilities. For instance, an increased rate of +1 additions could occur in any transaction involving DNA strand displacement synthesis, including recombination and DNA repair.

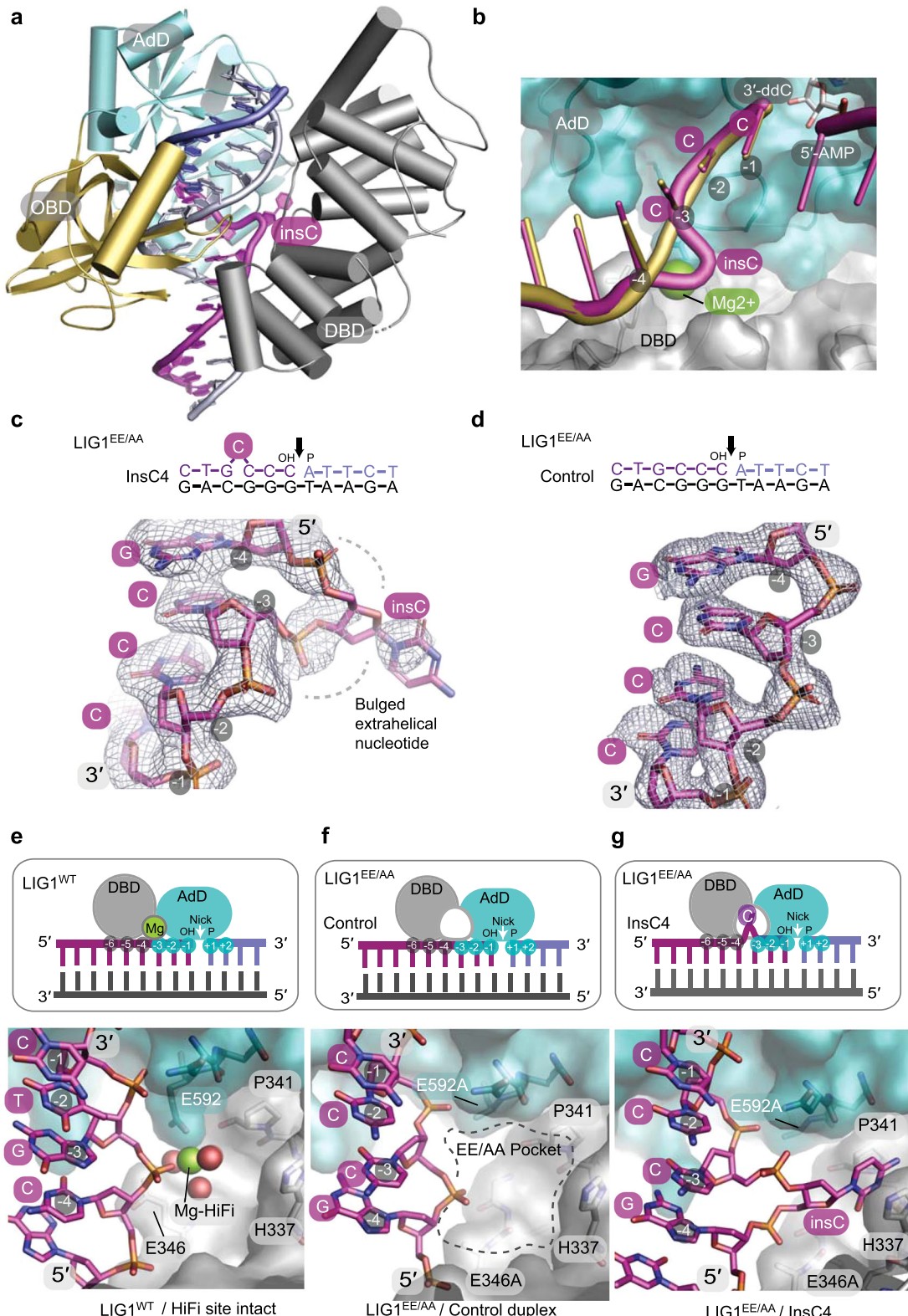

## Methods

**Yeast strain construction and growth analysis**. *S. cerevisiae* strains are isogenic derivatives of strain Δ|(−2)|−7B-YUNI300 (*MATa CAN1 his7-2 leu2Δ::kanMX ura3Δ trp1-289 ade2-1 lys2ΔGG2899-2900 agp1::URA3-OR1*)[45], and relevant genotypes are listed in Supplementary Table 1. The *cdc9-E206A-E443A-5FLAG:HphMX6* (*cdc9-EE/AA*) mutant strain was generated using a plasmid containing Cdc9 C-terminally tagged with 5×-FLAG and flanked by 600 bp of upstream and downstream sequences. Mutagenesis was performed using the QuikChange II Site-directed

Mutagenesis Kit (Agilent). Construction of diploids heterozygous for *MSH2/msh2::LEU2*, *MSH6/msh6::LEU2*, *MSH3/msh3::LEU2*, or *RAD27/rad27::LEU2* was performed by deletion replacement of one copy of each of the respective genes using the pUG73 plasmid[46]. Transformants were verified by marker selection and PCR analysis, as were the haploids derived from tetrad dissections. Yeast dissection plates were photographed after 3-day growth on rich (YPDA) medium at 30 °C.

Strains were grown in YPDA medium (1% yeast extract, 2% bacto-peptone, 250 mg l[−1] adenine, 2% dextrose, 2% agar for plates) at 30 °C. Doubling time (*D*t)

**Fig. 6 Molecular basis of mutagenic ligation by LIG1$^{EE/AA}$. a** LIG1$^{EE/AA}$ X-ray structure in complex with +1 nucleotide insertion DNA (insC4). The DBD (gray), AdD (teal), and OBD (gold) domains encircle a bulged nicked DNA substrate (bulged 3′-OH strand, magenta; 5′-P strand, blue; continuous strand, gray). **b** Structural overlay of LIG1$^{WT}$-DNA (PDB 6P09)[14] and LIG1$^{EE/AA}$-bulged DNA complexes shows that the backbone path of the bulged DNA (magenta) is distorted at the HiFi Mg$^{2+}$-binding pocket. LIG1$^{WT}$ binds the HiFi Mg$^{2+}$ (green) at the juncture between the 3′-OH of the upstream DNA (gold), AdD (teal), and DBD (gray). **c** Omit $F_o$–$F_c$ electron density contoured at 2$\sigma$ displayed for the bulged 3′-OH strand bound in the LIG1$^{EE/AA}$-bulged DNA complexes. **d**, Omit $F_o$–$F_c$ electron density contoured at 2$\sigma$ and displayed for the unbulged upstream 3′-OH strand bound in the LIG1$^{EE/AA}$-unbulged DNA complex. **e–g** Cartoon representations (top panels) and surface-filled representation of X-ray structures (bottom panels) depict protein–DNA contacts at the HiFi metal-binding sites of the LIG1$^{WT}$-DNA (**e**, PDB 6P09), LIG1$^{EE/AA}$-unbulged DNA (**f**), and LIG1$^{EE/AA}$-bulged insC4 DNA (**g**) complexes. The HiFi Mg$^{2+}$ bridging the nt −3 and nt −4 nucleotides of the 3′-OH strand relative to the nick site. The AdD binds across the broken DNA strands with the nick positioned over the active site and makes contacts with nt +2 to nt −3 while the DBD contacts nt −4 to nt −6. **f**, Removal of the HiFi Mg$^{2+}$ ligands (E592A/E346A) results in a cavity (EE/AA pocket). **g** The bulged extrahelical nucleotide (the fourth cytosine upstream of the DNA nick) in the LIG1$^{EE/AA}$-bulged DNA complex occupies the EE/AA pocket created by removal of the HiFi Mg$^{2+}$-binding site. Source data for panels are provided as a Source data file.

was measured for logarithmically growing cultures using between 4 and 12 replicates for each experiment. Data are displayed as the mean $D_t$ +/− standard deviation (SD). Microscopy was performed using cultures grown in rich medium at 30 °C to mid-logarithmic phase. Live cells were imaged with a Leitz Diaplan microscope combined with a Zeiss AxioCam MRm Rev.3 camera.

**Spontaneous mutation rate and sequencing analysis**. Mutation rate analysis was performed in strains containing the *URA3* reporter gene placed in orientation 1 adjacent to an efficient, early-firing replication origin, *ARS306*. In orientation 1, the leading and lagging strand of *URA3* have been defined in studies using mutator variants of DNA Pols δ and ε[33–38]. The coding sequence of the *URA3* reporter gene is displayed. Mutation rates and 95% confidence intervals were determined by measuring fluctuation analysis as described[47]. Because the *cdc9-EE/AA msh2Δ* and *cdc9-EE/AA rad27Δ* haploid strains were highly mutable, we sporulated diploid strains that were heterozygous for *msh2Δ* or *rad27Δ* and used freshly dissected independent spore colonies to measure the spontaneous mutation rate, thereby minimizing the number of generations during which mutations could rapidly accumulate and modulate mutation rates. The *ura3* gene from single, independent 5-FOA-resistant (5-FOA$^R$) colonies was PCR-amplified and sequenced. Specific mutation rates were calculated by multiplying the fraction of that mutation type by the total mutation rate for each strain.

**Statistical analysis**. All statistical analysis was performed using GraphPad Prism 8. *P* value determination for doubling time measurements was performed using the unpaired Student's *t* test, two tailed. Statistical analysis of overall mutation rate comparisons was performed using the one-sided nonparametric Mann–Whitney test.

**Mutagenesis, protein expression, and purification**. LIG1$^{EE/AA}$ was generated using QuikChange site-directed mutagenesis (Stratagene). LIG1$^{WT}$ and LIG1$^{EE/AA}$ proteins (aa262–904) were expressed in *Escherichia coli* Rosetta 2 (DE3) cells as previously described[14]. Genes encoding the Cdc9$^{WT}$ and Cdc9$^{EE/AA}$ proteins (aa112–754) were cloned into pET28M expression vector using NotI and BamHI restriction sites. The Cdc9 protein was expressed as an N-terminal His-tagged protein in *E. coli* Rosetta 2 (DE3) cells (Novagen). Cell cultures were grown at 37 °C in Terrific Broth supplemented with kanamycin (50 μg ml$^{-1}$) and chloramphenicol (34 ng ml$^{-1}$) until $A_{600}$ reached 1, at which time 50 μM IPTG was added. Protein expression was carried out at 16 °C overnight. Cells were harvested by centrifugation (6240 rcf, 20 min). Cell pellet was suspended and lysed in 30 ml lysis buffer (50 mM Tris, pH 8.5, 500 mM NaCl, 10 mM imidazole, 0.1 g lysozyme/ 1 l pellet, 1 tablet Roche mini EDTA-free protease inhibitor cocktail) at 4 °C for 30 min, followed by sonication. The cell lysate was fractionated by centrifugation (30,220 rcf, 20 min, 4 °C). The soluble fraction was applied to Ni-NTA resins (5 ml packed volume), which has been equilibrated with 15 ml (50 mM Tris, pH 8.5, 500 mM NaCl, 10 mM imidazole). The column was washed with 100 ml (50 mM Tris, pH 8.5, 500 mM NaCl, 10 mM imidazole), 15 ml (50 mM Tris, pH 8.5, 500 mM NaCl, 30 mM imidazole) and the His-tagged protein was eluted in 15 ml (50 mM Tris, pH 7.5, 500 mM NaCl, 300 mM imidazole). The His-tag was removed by TEV protease at 4 °C overnight. The untagged protein was purified on HiLoad 16/600 Superdex 200 gel filtration column in buffer (25 mM Tris, pH 7.5, 150 mM NaCl, 1 mM TCEP, 0.1 mM EDTA), followed by HiTrap SP HP 5 ml cation exchange column (low salt buffer: 20 mM Tris 7.5, 0.2 mM EDTA, 1 mM TCEP; high salt buffer: 20 mM Tris 7.5, 1 M NaCl, 0.2 mM EDTA, 1 mM TCEP). The quality of the purified proteins was analyzed by sodium dodecyl sulfate–polyacrylamide gel electrophoresis. Freshly purified proteins were concentrated and used immediately in crystallization experiments. Proteins used in activity assays were stored in (25 mM Tris, pH 7.5, 150 mM NaCl, 20% glycerol) at −80 °C until use.

**Ligation of Insertion DNA (insC1–insC8) and Controls (1c–8c)**. Synthetic oligos were used to generate Insertion DNA (insC1–insC8) and Control DNA (1c–8c) (IDT) (Supplementary Table 10). Each ligation reaction (10 μl) containing DNA ligase protein (2 nM), Insertion DNA, or Control DNA (50 nM) (Fig. 5a and Supplementary Fig. 6) in 10 mM Tris-HCl, pH 7.5, 10 mM MgCl$_2$, 10 mM DTT, 1 mM ATP, and 15 mM NaCl was carried out at 25 °C for 20 min, followed by a 10-min heat inactivation in 20 μl formamide loading buffer (98% formamide, 2% EDTA). Ligation reactions were resolved on 20% TBE-Urea gels and the Cy5-labeled reaction products were visualized on a Typhoon scanner (GE Healthcare) and analyzed using ImageQuant TL.

**Ligation of Insertion DNA 4 (insC4) and Control DNA 4 (4c)**. LIG1 and Cdc9 ligation activity on insC4 and 4c was evaluated. Ligation reactions (10 μl) containing insC4 or 4c (50 nM) (Fig. 5a and Supplementary Table 10) and DNA ligase protein (0–5000 nM) in 10 mM Tris-HCl, pH 7.5, 10 mM MgCl$_2$, 10 mM DTT, 1 mM ATP, and 115 mM NaCl were carried out at 25 °C for 20 min, followed by a 10-min heat inactivation in 20 μl formamide loading buffer (98% formamide, 2% EDTA). Ligation reactions were resolved on 20% TBE-Urea gels and Cy5-labeled reaction products were visualized on a Typhoon scanner (GE Healthcare). The percentages of DNA substrates, reaction intermediates, and products were generated by ImageQuant TL.

**Crystallization and structure determination**. Crystals of LIG1$^{EE/AA}$-unbulged DNA complex were grown by hanging drop method. 1 μL of protein–DNA complex solution (17 mg ml$^{-1}$ LIG1$^{EE/AA}$ (aa262–904), nicked 18mer from annealing oligos 1, 3, and 4 (Supplementary Table 11) (1.5:1 DNA:protein molar ratio), 1 mM ATP, 1 mM MgCl$_2$, in 150 mM NaCl, 20 mM Tris-HCl, pH 7.5, and 1 mM TCEP) with an equal volume of precipitant solution (100 mM MES, pH 6, 100 mM lithium acetate, 12% (w/v) polyethylene glycol 3350) at 20 °C. Crystals grew within 24 h and were washed in cryoprotectant (20% PEG3350 and 20% glycerol in precipitant solution supplemented with 100 mM MgCl$_2$) and flash frozen in liquid nitrogen for data collection.

Crystals of LIG1$^{EE/AA}$-bulged DNA complex were grown by hanging drop method. 1 μL of protein-DNA complex solution (20 mg ml$^{-1}$ LIG1$^{EE/AA}$ (aa262–904), nicked 18mer from annealing oligos 1, 2, and 4 (Supplementary Table 10) (1.5:1 DNA:protein molar ratio), 1 mM ATP, 1 mM MgCl$_2$, in 150 mM NaCl, 20 mM Tris-HCl, pH 7.5, and 1 mM TCEP) with an equal volume of precipitant solution (100 mM MES, pH 6, 150 mM lithium acetate, 10% (w/v) polyethylene glycol 3350) at 20 °C. The crystallization drops were allowed to equilibrate overnight. Microcrystal seeds prepared from crystals of LIG1$^{EE/AA}$-unbulged DNA complex was streaked into the drops. Crystals appeared after ~60 days and were washed in cryoprotectant (20% PEG3350 and 20% glycerol in precipitant solution supplemented with 100 mM MgCl$_2$) and flash frozen in liquid nitrogen for data collection.

X-ray diffraction data was collected on Beamline 22-ID of the Advanced Photon Source at a wavelength of 1.000 Å. X-ray diffraction data were processed and scaled using the HKL2000 suite[48]. All structures were solved by molecular replacement using PDB entry 6P09[49] as a search model with PHASER[49]. Iterative rounds of model building in COOT[50] and refinement with PHENIX[51] were used to produce the final models.

**Reporting summary**. Further information on research design is available in the Nature Research Reporting Summary linked to this article.

## Data availability

A reporting summary for this article is available as a Supplementary Information file. Atomic coordinates and structure factors for the reported crystal structures have been deposited with the RCSB Protein Data Bank under accession numbers 7KR3 and 7KR4. The data supporting the findings of this study are available from the corresponding authors upon request. Source data are provided with this paper.

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

## Acknowledgements

X-ray diffraction data were collected at the Southeast Regional Collaborative Access Team (SER-CAT) 22-ID beamline at the Advanced Photon Source, Argonne National Laboratory. Use of the Advanced Photon Source was supported by the U. S. Department of Energy, Office of Science, Office of Basic Energy Sciences under Contract No. W-31-109-Eng-38. We thank Lars Pedersen of the NIEHS collaborative crystallography group and the Advanced Photon Source (APS) Southeast Regional Collaborative Access Team (SER-CAT) staff for assistance with crystallographic data collection. The pET28M vector was a kind gift from L. Pedersen. We thank Bill Copeland for microscope access and Dmitry Gordenin, Andrea Kaminski, and Lars Pedersen for critical reading of the manuscript. This work was supported by the intramural research program of the US

National Institutes of Health (NIH), National Institute of Environmental Health Sciences (NIEHS) grants 1Z01ES102765 to R.S.W. and Z01ES065070 to T.A.K.

## Author contributions

Conceptualization: J.S.W., P.P.T., R.S.W., T.A.K.; methodology: J.S.W., P.P.T., M.E.A., R.S.W.; investigation: J.S.W., P.P.T., M.E.A., J.A.R., R.S.W.; data analysis: J.S.W., P.P.T., M.E.A., J.A.R., R.S.W.; writing—original draft: J.S.W., P.P.T., R.S.W., T.A.K.; writing—review and editing: J.S.W., P.P.T., M.E.A., R.S.W., T.A.K.; supervision: R.S.W., T.A.K.

## Competing interests

The authors declare no competing interests.
