## [Peer Review File · Nature Communications]

REVIEWERS' COMMENTS

Reviewer #1 (Remarks to the Author):

This paper reports high quality work from leading investigators in the ligase and polymerase fields. The data, integrating genetic, biochemical, and structural methods, are clean and convincing. The structures of the Cdc9-EE-AA mutant nicely point to the loss of metal occupancy as the key factor in allowing accommodation of a bulged nucleoside in the 3'-OH strand.

I recommend publication after the requested textual revisions listed below.

1) The claim of the title that ligase "regulates" fidelity is an over-reach. The results here show that a Cdc9 mutant allele cause a loss of fidelity, and suggest why, but there is no evidence that Cdc9's activity is subject to actual regulation with respect to its impact on fidelity. For example, there is no evidence that occupancy of the metal site on wild-type Cdc9 is a regulated event. The authors should change the title accordingly, along the lines of "A mutation of DNA ligase I that affects the fidelity of DNA replication"

2) Abstract line 23 and p. 3, line 60: "mononucleotide run" is confusing usage. "Homo-oligomeric run" might be better. Or "homonucleotide run" as used on . 5, line 88.

3) line 55; should be "despite the importance of..." (delete "central" – a dubious usage, e.g., in that there is no "peripheral importance")

4) p. 4, lines 75 and 76: use of "robust" as a descriptor is not very meaningful. Better to say, on line 75, that the EEAA mutant spores germinated and grow out identically to the wild-type control. And on line 76, say that the mutant ligase displays "X%" sealing activity in vitro compared to the wild-type Cdc9.

5) p. 4, line 78 and p. 9, line 193: delete "strikingly" (grammatically incorrect use of adverb). Ditto "interestingly" on p. 8, line 163.

6) Fig. 1a legend should include a statement identifying the source organisms used in the alignment of ligase aa sequences.

7) Fig. 1a: the label "Sc_Cdc17" seems like a mistake. I suppose the authors are referring to S. pombe Cdc17 (DNA ligase, which should be labeled Sp_Cdc17) and not S. cerevisiae Cdc17 (catalytic subunit of Pol alpha-primase).

Reviewer #2 (Remarks to the Author):

DNA ligase 1 provides a vital function in cells as it ligates processed Okazaki fragments during lagging strand synthesis. Surprisingly, the fidelity of Ligase 1 (LIG1) has not been studied at a mechanistic level. This tour-de-force study combining genetic, structural biology and biochemical results, by a team of outstanding scientists, who have investigated the effects of a LIG1 variant in which two key glutamate residues have been substituted to alanines. These Glu residues are believed to be important for high-fidelity metal binding. Using the URA3 reporter gene assay, the authors first show that this LIG1 variant shows a slight increase in mutation frequency, which when combined with altered mismatch repair or FEN1 (Rad27) nuclease shows a synergistic increase in mutation frequency. The spectra of mutations show increased +1 insertion in homonucleotide runs. Further analysis suggest that these are occurring on the lagging strand. The authors then show in the absence of MSH2 and RAD27 yeast carrying the EE/AA LIG1 variant do not grow. Based on these data the authors present a working model for LIG1 fidelity and then present and biochemical and elegant structural data supporting this model. Specifically they biochemically that bringing a bulged C close to the ligatable nick diminished ligation by the WT LIG1, which was arrest at the 5-AMP state, but show that removal of the two EE residues in the variant allows ligation of the bulge. Furthermore control experiments show that both DNA ligases

have relative equal activity on a nicked substrate. They further show structural that EE to AA substitutions creates a pocket that can accommodate a flipped out residue and allow for replication slippage and subsequent +1 insertional frameshift. This is an excellent study that provides an important new and innovative data providing mechanistic insight into LIG1 fidelity that should have a long and lasting impact on the field. There are very few problems with this work, the following issue needs clarification.

Figure 1, Panel C shows the accumulation of mutations on the coding strand which is the leading strand, but then the authors argue that the mutations were arising on the lagging strand in Figure 2. The logic on lines 148-150 makes sense, but since the authors used PCR fragments to sequence the mutations in the URA3, it is not clear how they can specifically assign specific mutations to one strand or the other. Please clearly describe in the methods. It may be confusing to reader to show mutational spectra on the leading strand in Figure 1 and in Supplemental figures, but then argue that these arose on the lagging strand. Would redrawing these on the opposite strand make more sense. Is there a specific Pol delta mutant that could be used to prove this hypothesis?

Despite the beautiful renderings in Figure 6, an added supplemental movie that shows the WT structure morphing into the variant structure and subsequent accommodation of the bulged C would be very helpful to the reader.

Reviewer #3 (Remarks to the Author):

The manuscript by Williams et al, makes an excellent case for the need for precise, high-fidelity ligation during Okazaki fragment maturation in order to maintain the fidelity of the genome. Comparing ligation efficiency of the wild-type and a mutant variant which exhibits low fidelity, the authors show that a mutation in the high-fidelity metal binding site of LIG1 allows for promiscuous ligation and introduction of insertion mutations. This work is a natural extension of their previous work characterizing the metal binding site of human LIG1. Moving into the yeast model, the authors have relied on their robust reporter assays to show the importance of the high-affinity metal binding site in coordinating efficient ligation. When the metal binding site was altered, the mutant variant displayed an increased amount of single base addition mutations compared to the wild-type strain, with mutations clustered mostly in GC runs. Deletions of both flap processing enzyme, Rad27 and proteins from the mismatch repair pathway (MSH2) accentuated the insertions in the LIG1 mutant variant strain. Biochemical assays reveal that the mutant variant was more efficient at ligating bulged (+1) substrates compared to the wild type ligase. Finally, the authors provided a crystal structure for the mutant variant and showed how the bulge in the substrate is accommodated in a specific pocket that would have normally been occupied by the Mg²⁺. The genetic experiments clearly show an in vivo impact of altering the high-fidelity metal binding site. These findings are ably supported by both biochemical experiments and the crystal structure of the mutant ligase. The data presented are of high quality and the authors have provided good explanations for their observations. Overall, the manuscript provides an excellent model for the high-fidelity metal binding site and the role it plays in maintenance of genome stability.

Minor comments:

1. The authors considered +1 additions to be a result of either deficiency in MMR or caused during flap processing by Rad27 (and/or Dna2). Would the authors expect higher rates of +1 insertions in an exonuclease deficient strain of DNA polymerase delta?
2. Ligation assays: Supplementary Figure 6: The ligation efficiency of the WT-LIG1 seems to vary drastically on the substrates. For example, almost 90% of the substrate is ligated in substrates 1c and 2c, whereas ~ only 10% ligation of other substrates (3c, 6c, 7c) are observed. Since similar concentration of enzyme was present in each reaction, one would expect the ligation efficiency to be similar in each substrate. Could substrate breathing be a reason for this varying ligation efficiency? Similarly, for Figure 5, could the bulge destabilize the 3' annealing and thus not provide a stable nick for the ligase activity?
3. One can expect the bulge to be also present on the downstream primer (in case of equilibrating flaps). The bulge would probably not be accommodated in a similar manner as shown in the crystal structure. How would this impact ligation efficiency?

REVIEWERS' COMMENTS

Reviewer #1:

This paper reports high quality work from leading investigators in the ligase and polymerase fields. The data, integrating genetic, biochemical, and structural methods, are clean and convincing. The structures of the Cdc9-EE-AA mutant nicely point to the loss of metal occupancy as the key factor in allowing accommodation of a bulged nucleoside in the 3'-OH strand.

Response: We thank the reviewer for their positive comments on the manuscript.

I recommend publication after the requested textual revisions listed below.

1) The claim of the title that ligase “regulates” fidelity is an over-reach. The results here show that a Cdc9 mutant allele cause a loss of fidelity, and suggest why, but there is no evidence that Cdc9’s activity is subject to actual regulation with respect to its impact on fidelity. For example, there is no evidence that occupancy of the metal site on wild-type Cdc9 is a regulated event. The authors should change the title accordingly, along the lines of “A mutation of DNA ligase I that affects the fidelity of DNA replication”

Response: Thank you, we agree. We have changed the title to:
“High-fidelity DNA ligation enforces accurate Okazaki fragment maturation during DNA replication”

2) Abstract line 23 and p. 3, line 60: “mononucleotide run” is confusing usage. “Homo-oligomeric run” might be better. Or “homonucleotide run” as used on . 5, line 88.

Response: We agree and have changed the line to read: “... single base insertion mutations in homonucleotide runs.”

3) line 55; should be “despite the importance of...” (delete “central” – a dubious usage, e.g., in that there is no “peripheral importance”)

Response: We removed the word ‘central’ from the sentence.

4) p. 4, lines 75 and 76: use of “robust” as a descriptor is not very meaningful. Better to say, on line 75, that the EEAA mutant spores germinated and grow out identically to the wild-type control.

Response: We changed the text to include these changes. Lines 75-76 now read: “Tetrad dissection reveals that spore colonies harboring the mutant *cdc9-EE/AA* ligase germinated and grew to a colony size similar to a wild type control (Supplementary Fig. 1).

And on line 76, say that the mutant ligase displays “X%” sealing activity in vitro compared to the wild-type Cdc9.

Response: The ligation activities of mutant and WT enzyme are discussed later in the text, so we removed this sentence.

5) p. 4, line 78 and p. 9, line 193: delete "strikingly" (grammatically incorrect use of adverb). Ditto "interestingly" on p. 8, line 163.

Response: We have removed 'strikingly' and 'interestingly'.

6) Fig. 1a legend should include a statement identifying the source organisms used in the alignment of ligase aa sequences.

Response: We now include a description of the source organisms used in the ligase amino acid sequence alignments in the legend to Fig. 1a. The legend reads: "Sc, *Saccharomyces cerevisiae*; Hs, *Homo sapiens*; Mm, *Mus musculus*; Bt, *Bacillus thuringiensis*; Xl, *Xenopus laevis*; Dm, *Drosophila melanogaster*; Sp, *Schizosaccharomyces pombe*".

7) Fig. 1a: the label "Sc_Cdc17" seems like a mistake. I suppose the authors are referring to *S. pombe* Cdc17 (DNA ligase, which should be labeled Sp_Cdc17) and not *S. cerevisiae* Cdc17 (catalytic subunit of Pol alpha-primase).

Response: Thank you, this is the *S. pombe* protein; Sp_Cdc17.

Reviewer #2:

DNA ligase 1 provides a vital function in cells as it ligates processed Okazaki fragments during lagging strand synthesis. Surprisingly, the fidelity of Ligase 1 (LIG1) has not been studied at a mechanistic level. This tour-de-force study combining genetic, structural biology and biochemical results, by a team of outstanding scientists, who have investigated the effects of a LIG1 variant in which two key glutamate residues have been substituted to alanines. These Glu residues are believed to be important for high-fidelity metal binding. Using the URA3 reporter gene assay, the authors first show that this LIG1 variant shows a slight increase in mutation frequency, which when combined with altered mismatch repair or FEN1 (Rad27) nuclease shows a synergistic increase in mutation frequency. The spectra of mutations show increased +1 insertion in homonucleotide runs. Further analysis suggest that these are occurring on the lagging strand. The authors then show in the absence of MSH2 and RAD27 yeast carrying the EE/AA LIG1 variant do not grow. Based on these data the authors present a working model for LIG1 fidelity and then present and biochemical and elegant structural data supporting this model. Specifically they biochemically that bringing a bulged C close to the ligatable nick diminished ligation by the WT LIG1, which was arrested at the 5-AMP state, but show that removal of the two EE residues in the variant allows ligation of the bulge. Furthermore control experiments show that both DNA ligases have relative equal activity on a nicked substrate. They further show structural that EE to AA substitutions creates a pocket that can accommodate a flipped out residue and allow for replication slippage and subsequent +1 insertional frameshift. This is an excellent study that provides an important new and innovative data providing mechanistic insight into LIG1 fidelity that should have a long and lasting impact

on the field. There are very few problems with this work, the following issue needs clarification.

Response: We thank the reviewer for their positive comments on the manuscript and are happy to make the requested changes.

Figure 1, Panel C shows the accumulation of mutations on the coding strand which is the leading strand, but then the authors argue that the mutations were arising on the lagging strand in Figure 2. The logic on lines 148-150 makes sense, but since the authors used PCR fragments to sequence the mutations in the URA3, it is not clear how they can specifically assign specific mutations to one strand or the other. Please clearly describe in the methods. It may be confusing to reader to show mutational spectra on the leading strand in Figure 1 and in Supplemental figures, but then argue that these arose on the lagging strand. Would redrawing these on the opposite strand make more sense. Is there a specific Pol delta mutant that could be used to prove this hypothesis?

Response: We apologize for the confusion and appreciate the reviewer's feedback on this important issue. Therefore, we have taken this opportunity to clarify our *URA3* mutation spectra figures and explanations. The assignment of these mutations to the lagging strand is based on previous work using both mutation signatures (Pursell et al., *Science* (2007); Nick McElhinny et al., *Mol Cell* (2008)) and ribonucleotide incorporation (Lujan et al., *PLoS Genet* (2012); Williams et al., *Mol Cell* (2013); Lujan et al., *Mol Cell* (2013); Williams et al. *Nat Struct Mol Biol* (2015)) to define the leading and lagging strands for the *URA3* reporter gene in its position adjacent to the closest replication origin, *ARS306*. These studies were performed using mutator variants of DNA polymerases δ and ϵ that had been biochemically characterized as having specific mutation signatures and ribonucleotide incorporation propensities. We now include this more detailed description in the text on page 8 lines 150-152, and have also described our logic for assigning mutations to each strand in the Methods section on pages 14-15.

We display the sequence of the *URA3* coding strand in the mutation spectra as a matter of convention and to keep our analysis consistent with previous publications where this same reporter system was utilized. Because the +1 insertion mutations that arise in the low-fidelity *cdc9-EE/AA* mutant strain do not appear to have strict sequence specificity (Fig. 2c), it may be difficult to use a specific DNA polymerase delta mutant or other genetic trick to provide further support of the hypothesis that they are arising during lagging strand synthesis.

Despite the beautiful renderings in Figure 6, an added supplemental movie that shows the WT structure morphing into the variant structure and subsequent accommodation of the bulged C would be very helpful to the reader.

Response: We thank the reviewer for this excellent feedback and have added a supplemental movie (Supplementary Movie 1) that shows a structural morph between the un-bulged and bulged DNA-bound states of mutant *LIG1^{EE/AA}*-DNA complexes. The bulged extrahelical nucleotide in the insC complex flips into a pocket created by alanine mutations in the Add (turquoise) and DBD (grey) domains.

Reviewer #3:

The manuscript by Williams et al, makes an excellent case for the need for precise, high-fidelity ligation during Okazaki fragment maturation in order to maintain the fidelity of the genome. Comparing ligation efficiency of the wild-type and a mutant variant which exhibits low fidelity, the authors show that a mutation in the high-fidelity metal binding site of LIG1 allows for promiscuous ligation and introduction of insertion mutations. This work is a natural extension of their previous work characterizing the metal binding site of human LIG1. Moving into the yeast model, the authors have relied on their robust reporter assays to show the importance of the high-affinity metal binding site in coordinating efficient ligation. When the metal binding site was altered, the mutant variant displayed an increased amount of single base addition mutations compared to the wild-type strain, with mutations clustered mostly in GC runs. Deletions of both flap processing enzyme, Rad27 and proteins from the mismatch repair pathway (MSH2) accentuated the insertions in the LIG1 mutant variant strain. Biochemical assays reveal that the mutant variant was more efficient at ligating bulged (+1) substrates compared to the wild type ligase. Finally, the authors provided a crystal structure for the mutant variant and showed how the bulge in the substrate is accommodated in a specific pocket that would have normally been occupied by the Mg²⁺. The genetic experiments clearly show an in vivo impact of altering the high-fidelity metal binding site. These findings are ably supported by both biochemical experiments and the crystal structure of the mutant ligase. The data presented are of high quality and the authors have provided good explanations for their observations. Overall, the manuscript provides an excellent model for the high-fidelity metal binding site and the role it plays in maintenance of genome stability.

Response: We thank the reviewer for their positive comments on the manuscript.

Minor comments:

1. The authors considered +1 additions to be a result of either deficiency in MMR or caused during flap processing by Rad27 (and/or Dna2). Would the authors expect higher rates of +1 insertions in an exonuclease deficient strain of DNA polymerase delta?

Response: We do anticipate that the rate of +1 insertion mutations may be higher in an exonuclease-deficient DNA polymerase delta mutant strain. This is based on the possibility that DNA polymerase delta proofreads +1 insertions that arise during strand displacement synthesis. If +1 insertions that are the result of strand slippage and incorporation of an extra nucleotide by DNA polymerase delta fail to be proofread, one would expect an elevated rate of +1 insertion mutations in a yeast strain with an exonuclease-deficient DNA Pol δ (e.g. *pol3-5DV*). This is an area of investigation that we are intrigued by and currently investigating. Interestingly, very little is known about the efficiency of 3'-exonucleolytic proofreading by DNA polymerase delta of +1 insertion mutations that arise during strand displacement synthesis, and our hope is that these current and future studies may shed some light on this issue. We have included the following sentence on page 13 line 271 in the Discussion to acknowledge the possibility that Pol δ can proofread +1 insertion mutations that arise during strand displacement synthesis: "These results suggest that Pol δ makes +1 slippage mutations frequently during strand displacement synthesis,

and we are currently testing whether the 3'-exonuclease activity of Pol δ is capable of proofreading such replication errors.”.

2. Ligation assays: Supplementary Figure 6: The ligation efficiency of the WT-LIG1 seems to vary drastically on the substrates. For example, almost 90% of the substrate is ligated in substrates 1c and 2c, whereas ~ only 10% ligation of other substrates (3c, 6c, 7c) are observed. Since similar concentration of enzyme was present in each reaction, one would expect the ligation efficiency to be similar in each substrate.

Response: Thank you for this comment. In this experiment, we translated the nick position on a fixed sequence scaffold. This approach changes both the register and sequence context, so the overall activity is modulated by local nick position, the bulge, and sequence context. It is for this reason that we employed enzyme titrations to more quantitatively examine Cdc9 and LIG1 WT and EE/AA variants on the insC4 substrate and controls (Fig. 5f).

Could substrate breathing be a reason for this varying ligation efficiency? Similarly, for Figure 5, could the bulge destabilize the 3' annealing and thus not provide a stable nick for the ligase activity?

3. One can expect the bulge to be also present on the downstream primer (in case of equilibrating flaps). The bulge would probably not be accommodated in a similar manner as shown in the crystal structure. How would this impact ligation efficiency?

Response: Thank you, these are great points. Indeed, the bulge could be accommodated in any of 4 registers, including a 3'-flap conformation that is presumably non-ligatable. In fact, there is a significant (~ 20%) fraction of the insC4 substrate that remains unligatable by the Cdc9-EE/AA protein, even at our highest enzyme concentrations, suggesting that variable conformations in annealing of the 3' bulge also impact ligation efficiency. We now comment on this in the text.